# Multi-Hop Question Generation with Knowledge Graph-Enhanced Language Model

Zhenping Li [1,2,†] , Zhen Cao [3,†], Pengfei Li [3,†], Yong Zhong [1,2] and Shaobo Li [1,4,*]

1 Chengdu Institute of Computer Applications, Chinese Academy of Sciences, Chengdu 610041, China; zhongyong@casit.com.cn (Y.Z.)
2 School of Computer Science and Technology, University of Chinese Academy of Sciences, Beijing 100049, China
3 School of Electrical and Electronic Engineering, Nanyang Technological University, Singapore 639798, Singapore
4 Key Laboratory of Advanced Manufacturing Technology, Ministry of Education, Guizhou University, Guiyang 550025, China
* Correspondence: lishaobo@gzu.edu.cn
† These authors contributed equally to this work.

**Abstract:** The task of multi-hop question generation (QG) seeks to generate questions that require a complex reasoning process that spans multiple sentences and answers. Beyond the conventional challenges of what to ask and how to ask, multi-hop QG necessitates sophisticated reasoning from dispersed evidence across multiple sentences. To address these challenges, a knowledge graph-enhanced language model (KGEL) has been developed to imitate human reasoning for multi-hop questions.The initial step in KGEL involves encoding the input sentence with a pre-trained GPT-2 language model to obtain a comprehensive semantic context representation. Next, a knowledge graph is constructed using the entities identified within the context. The critical information in the graph that is related to the answer is then utilized to update the context representations through an answer-aware graph attention network (GAT). Finally, the multi-head attention generation module (MHAG) is performed over the updated latent representations of the context to generate coherent questions. Human evaluations demonstrate that KGEL generates more logical and fluent multi-hop questions compared to GPT-2. Furthermore, KGEL outperforms five prominent baselines in automatic evaluations, with a BLEU-4 score that is 27% higher than that of GPT-2.

**Keywords:** multi-hop question generation; graph neural network; natural language processing; reasoning chain

## 1. Introduction

Question generation (QG) endeavors to automatically generate high-quality questions through reasoning and inference from context and answers. QG can provide a wealth of training data for question answering (QA) tasks [1] or serve as a starting point for dialogue systems [2]. Much of the research in text-based QG focuses on single-hop reasoning (e.g., the SQuAD dataset) [3–6], where each question–answer pair is usually derived from a single sentence. For example, in the SQuAD dataset, a question such as "In what country is Normandy located?" would be generated after reading the sentence "The Normans were the people who in the 10th and 11th centuries gave their name to Normandy, a region in France". This type of single-hop QG, although useful, is considered limited in terms of its ability to mimic human-like intelligence. A human, on the other hand, can easily ask insightful questions after reading a book with a complex narrative. For instance, one might ask "What caused Voldemort to become so evil, or is his villainy 'inherent'?" after reading the Harry Potter series. This type of multi-hop QG constitutes a significant challenge that must be addressed in the pursuit of advanced AI. To address the multi-hop

question generation task, we adopt a straightforward learning paradigm that consists of two phases [7]. The first phase involves learning what to ask by identifying and integrating crucial evidence in sentences. The second phase involves learning how to ask by generating logical and fluent questions based on the identified evidence.

Our model faces three main challenges in this learning paradigm. The first challenge is to effectively identify and integrate crucial evidence from sentences for learning what to ask. The copy mechanism [6] has been widely used for finding words related to questions in context. Pan et al. [8] addressed this challenge by fusing document-level and graph-level representations through an attention-based gate graph neural network to select answer-related evidence information. Chauhan et al. [9] employed multi-task learning with an auxiliary loss to predict sentence-level evidence. To aggregate potential evidence related to questions, Su et al. [10] leveraged an answer-aware dynamic entity graph constructed from mentioned entities in the answer and input paragraphs.

The second challenge in the multi-hop QG task is to perform complex reasoning over scattered evidence and form a reasoning chain that connects the answer to the question. To address this challenge, several approaches have been proposed, such as the dynamic multi-hop reasoning-based model proposed by Ji et al. [11], which collects evidence using relational paths. The integration of external commonsense knowledge was also explored as a means of enhancing reasoning ability [12,13]. The third challenge is to generate fluent and logically consistent questions. Several pre-trained language models, such as GPT-2 [14], BART [15], and UniLM [16], have achieved success in various language generation tasks. However, these models often struggle with multi-hop reasoning, as they are designed with a self-attention module that only considers single-hop (or obvious) relationships among words, leading to shallow generation that fails to reflect human-like multi-hop reasoning processes.

It is worth noting that some research has addressed one or two of the three challenges, but there is currently no single approach that addresses all three simultaneously. For instance, RNN-based QG models [3,6] are prone to generating questions with grammatical errors, and pre-trained models such as GPT-2 [14] often generate non-logical or irrelevant questions, as demonstrated in Figure 1.

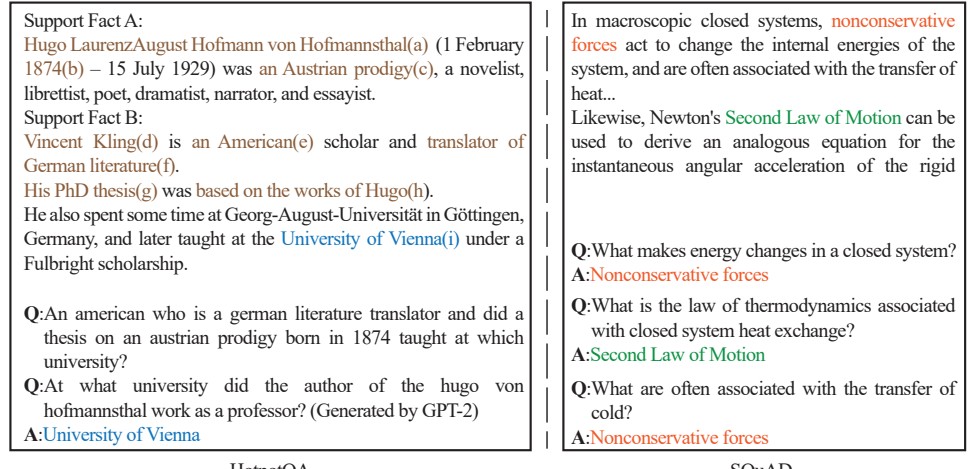

**Figure 1.** Examples of multi-hop QG (**left**) and single-hop QG (**right**). Multi-hop QG needs to perform reasoning from evidence (a) to (i), whereas single-hop QG generates questions from one sentence that explicitly states the answer. The question generated by GPT-2 (**left**) is irrelevant to the context.

This paper proposes a novel knowledge graph-enhanced language model to cope with the challenges noted above by a straightforward learning paradigm "learn what to ask and learn how to ask". Our model split the learning paradigm into three steps.

- The utilization of a GPT-2 model for encoding the context and answer allows for better semantic understanding, leveraging the pre-trained knowledge from its large-scale corpus.
- The bidirectional attention mechanism in the answer-aware GAT enables the model to dynamically capture the interactions between the answer and the knowledge graph, thus leading to a more comprehensive and accurate aggregation of the relevant information. Furthermore, the Graph2Context encoding allows the context representations to be updated with the knowledge derived from the knowledge graph, providing the model with a better semantic understanding of the context and the answer. These innovations enable the model to have a more thorough understanding of the context and answer, and, therefore, generate higher quality questions.
- Additionally, the multi-head attention generation module adopts a multi-head attention mechanism to capture the relationships among the latent representations, resulting in a more comprehensive understanding of the context. Furthermore, the module generates questions based on the enhanced understanding of the context, ensuring the fluency and logicality of the generated questions. The technical details of the module and the entire approach are thoroughly described in Section 2 of the paper.

Neural networks are commonly regarded as black boxes that require experimental validation to justify their effectiveness. Theoretical and mathematical explanations are lacking in this paper, but our results demonstrate that the framework we envisioned can improve performance.

To address, identify, and integrate crucial evidence from sentences and complex reasoning over scattered evidence, we propose a novel approach that combines entity recognition models and graph neural networks to enhance evidence discovery and sorting. In line with human cognition, we hypothesize that evidence is often expressed as a concrete noun. Therefore, upon identifying an entity, we expect to uncover evidence that is pertinent to the problem. Our graph neural network model is designed to function like a chain of evidence (it is the answer-aware GAT in Figure 2), connecting information among entities. By adopting a human-like reasoning process, we anticipate the model to identify key evidence about answers (it is the bi-attention in Figure 2) and integrate them to capture relevant information more accurately. To generate fluency, we choose a pre-trained language model to read and generate the language. Our proposed framework is founded on a straightforward approach that comprises three fundamental steps. First, a pre-trained model is leveraged to comprehend the text. Second, a graph neural network is employed to enhance the interaction of entities within the text. Finally, the information gleaned from the preceding two steps is amalgamated and organized. It is important to note that our framework is not constrained to any specific model.

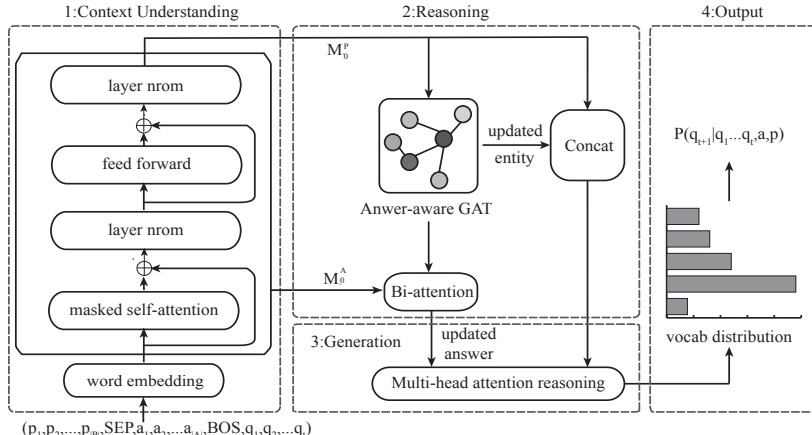

**Figure 2.** Framework of KGEL. It consists of three major modules: a context understanding module, a reasoning module, and a generation module.

Our contributions are summarized as follows: The proposed method for multi-hop question generation offers a novel approach that mirrors the human multi-hop questioning process. Our approach is composed of three phases: (1) context understanding through the utilization of a pre-trained GPT-2 language model, (2) integration of information and reasoning through a knowledge graph and an answer-aware dynamic GAT, and (3) QG through a multi-head attention generation module with enhanced latent representations. The efficacy of the proposed method is demonstrated through experiments conducted on the HotpotQA dataset, resulting in substantial advancements in automatic evaluation metrics, such as BLEU and ROUGE-L, as well as human evaluations, including fluency, answerability, and completeness.

## 2. Related Work

The field of question generation (QG) has undergone several stages of evolution, each characterized by the type of features leveraged. The first stage involved the use of rule-based approaches [17,18] that relied on human-designed features and utilized semantic information from the context to generate questions. The second stage marked the emergence of sequence-to-sequence models, which utilized neural encoder–decoder architectures to automatically learn semantic information from large datasets. This stage was primarily focused on single-hop QG, with works such as [3] designing the first neural encoder–decoder model for QG without considering answer information. This was followed by several other neural network-based QG approaches that utilized different types of encoders and decoders [4,6,19]. These works explored the relationship between the answer and context for QA problems and performed single-hop reasoning using different attention mechanisms [4,20]. The third stage has been marked by the emergence of multi-hop QG. Chauhan [9] utilized multi-task and reinforcement learning to enhance the performance of QG models. Additionally, graph neural network frameworks have been explored to improve reasoning ability in multi-hop QA tasks, such as graph convolutional networks [21] and graph attention networks [22], with recent works showing promising results. Examples include [23], which proposed a dynamically fused graph network for multi-hop QA on the HotpotQA dataset, and [24], which proposed an entity-GAT method to reason across multiple documents for multi-hop QA on the WIKIHOP dataset [25].

## 3. Methodology

The proposed KGEL model comprises three distinct components, which are illustrated in Figure 2.

**Reading Component**. The contextual words are processed by a pre-trained GPT-2 language model, which helps to obtain enhanced semantic representations of the context.

**Reasoning Component**. The answer-aware GAT integrates the entity information from the knowledge graph into the context, updating the context representations. Furthermore, the bi-attention mechanism updates the answer representations with the information from the knowledge graph.

**Generation Component**. The enhanced context representations obtained from the previous steps are analyzed using a multi-head self-attention module [26] to generate the question.

### 3.1. Context Understanding

Let $P$, $Q$, $A$ denote context passage, question to be generated, and answer, respectively; $P = (p_1, p_2, p_3, \ldots, p_{|P|})$, $A = (a_1, a_2, a_3, \ldots, a_{|A|})$, and $Q = (q_1, q_2, q_3, \ldots, q_{|Q|})$ are sequences of an arbitrary length, where $p_i$, $a_i$, and $q_i$ denote a word at position $i$ of the passage, answer, and question, respectively. In addition, $|P|$, $|A|$, and $|Q|$ are the number of words in $P$, $A$, and $Q$, respectively. Inspired by Zhao et al. [6], $A$ is appended to $P$, separated by a special $SEP$ token. The concatenated sequence $[P, SEP, A]$ is fed into GPT-2 to obtain initial contextual representation matrices for $P$ and $A$, illustrated as

$$M_0^P, M_0^A = \text{GPT2}([P, \text{SEP}, A]), \tag{1}$$

where $M_0^P \in \mathbb{R}^{|P| \times d_G}$ and $M_0^A \in \mathbb{R}^{|A| \times d_G}$, and $d_G$ is the dimension of the hidden state of GPT-2.

The reasoning component of the proposed KGEL model is depicted in Figure 3. The aim of this component is to enhance the representations of both the context and answer by incorporating entity information from the knowledge graph. This is achieved through the use of an answer-aware graph attention network (GAT) to update the context representations and a bi-attention mechanism to update the answer representations. The workings of the reasoning component are detailed in the subsequent sub-sections.

**Knowledge Graph Construction and Entity Encoding**. A BERT-based NER model is employed to identify entities in the context and to annotate their positions for ease of extracting their corresponding embeddings from the GPT-2 embedding sequence. The resulting knowledge graph is constructed with the entities in the context serving as its nodes and edges being established between entities within the same paragraphs and between entities in each paragraph and those in the title.

With the context passage encoding $M_0^P$ obtained from the GPT-2 encoder, a binary-valued context-entity mapping matrix $M^m$ is used to obtain entity encoding, where $M_{i,j}^m = 1$ if the corresponding position is within the span of the $j$th entity in the $i$th context, otherwise $M_{i,j}^m = 0$. Context-to-graph operation, illustrated in the upper left corner of Figure 3, aims to obtain the entity embedding matrix $M_0^E$, which is calculated by mean–max pooling over corresponding passage encoding span after the mapping operation:

$$M_0^E = \text{Mean} - \text{maxpooling}(M_0^P M^m), M_0^E \in \mathbb{R}^{|E| \times 2d_G}, \tag{2}$$

where $d_G$ is the hidden size of GPT-2, and $|E|$ is the number of entities extracted from the input passage.

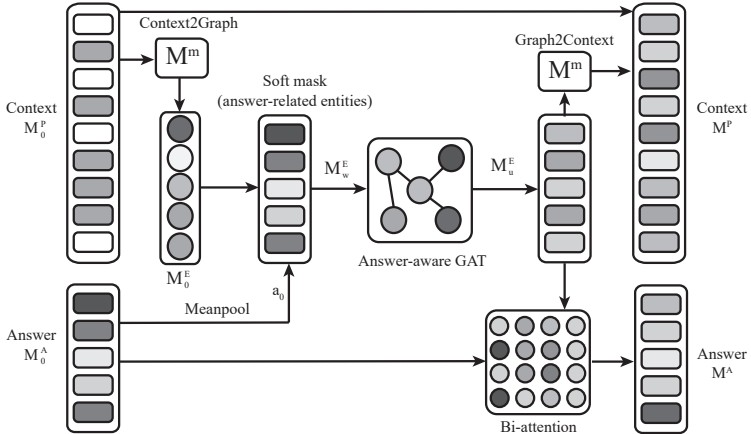

**Figure 3.** The illustration of the reasoning module. We implement a dynamic procedure of Context2Graph and Graph2Context using a mapping matrix M. The entity representation is updated using an answer-aware GAT, and the answer representation is updated through a bi-attention mechanism. A data flow example is shown in Appendix A.

**Answer-aware GAT**. The answer-aware GAT is proposed to gather scattered answer-related entity information and mimic a human's ability to process information in each sequential manner to form a reasoning chain. Inspired by Qiu et al. [23], after obtaining entities' embedding matrix $M_0^E$, a graph neural network is utilized to propagate answer-aware node information between answer related entities. To achieve this, a soft mask mechanism is applied to weigh the nodes considering the answer information as described

in Figure 3. Answer representation is obtained with a mean pooling operation from the initial contextual representation of $A$:

$$a_0 = \text{Meanpooling}(M_0^A). \tag{3}$$

To constrain the model to focus on answer-related entities, the entities of the knowledge graph are weighted by a soft mask calculated with an attention mechanism that explicitly model the correlation between entities and the answer $r$, which is formulated as:

$$\gamma_i = a_0 V^E M_0^E[i], i \in (1, |E|), \tag{4}$$
$$r = \sigma([\gamma_1, \ldots, \gamma_{|E|}]), \tag{5}$$

where $V^E$, $|E|$, and $\sigma$ are the linear projection matrix, the number of entities, and a sigmoid function. Therefore, the weighted answer-related entity embedding matrix is

$$M_w^E = sM_0^E. \tag{6}$$

Thus the information related to the answer is promoted while irrelevant information is demoted with the weighted graph. Inspired by GAT [22], the attention attn of two entities in the weighted graph is formulated as

$$h_i^E = W^E M_w^E[i], \tag{7}$$
$$\beta_{i,j}^E = \text{LeakyReLU}(W_L^E[h_i^E, h_j^E]), \tag{8}$$
$$\text{attn}_{i,j}^E = \frac{\exp(\beta_{i,j}^N)}{\sum_j \exp(\beta_{i,j}^N)}, \tag{9}$$

where $W^E$ and $W_L^E$ are linear projection parameters; LeakyReLU is the activation function. The $i$th row of attn represents the proportion of information that will be assigned to the neighbors of the $i$th entity. The $i$th node sums over its column in the proposed answer-aware GAT, which forms an updated entity state $e_i^{uE}$ containing all information received from the neighbors such that

$$e_i^{uE} = \text{ReLU}(\sum_{j \in S_i} \text{attn}_{j,i}^E h_j^E), \tag{10}$$

where $S_i$ is the neighbor entity set for the $i$th entity. Thus, the updated entity embedding matrix is given:

$$M_u^E = [e_i^{uE}, \ldots, e_{|E|}^{uE}], \tag{11}$$

where $|E|$ is the number of the entity. The details of data processing in answer-aware GAT are covered in Appendix A.

**Graph to Answer Encoding**. With the modelling of answer-aware entities, the reasoning module updates its answer representation $M^A$ via a Biattn [27]:

$$M^A = \text{Biattn}(M_0^a, M_u^E), \tag{12}$$

where Biattn is a bi-attention function, which can attend to both sources. The present study refers to $M_0^a$ as the embedding representation of the answer paragraph. We utilize $M_u^E$ as the embedding representation of all entities and facilitate information exchange between $M_0^a$ and $M_u^E$ via the bi-attention mechanism. This augmentation leads to an improved associativity between the answer and all entities. By incorporating this processing step, we anticipate that the entities closely related to the answer will receive heightened attention.

**Graph to Context Encoding**. With answer-aware graph analysis, the context can be reviewed to gain a better understanding of multi-hop QG by considering the interaction between answers and answer related entities.

Then, a Graph2Context encoding module is developed to update the enhanced answer-aware entities' embedding (achieved by answer-aware GAT) to the context representation, using the same mapping matrix $M^m$ in the graph construction section. The original context embeddings in $M_0^P$ are concatenated and fused with corresponding updated entity embeddings with a linear layer given as

$$M^P = \text{ConcatLinear}(M_0^P, M^E M^m). \tag{13}$$

*3.2. Question Generation*

The preceding steps establish a reasoning chain from the answer to answer-related entities, feeding the information of the chain into the context embedding. This aggregation of information from the context perspective aims to result in the generation of a logical and fluent question. The utilization of self-attention is motivated by its capability to capture the interactions among the words in a sentence. Inspired by the multi-head self-attention mechanism introduced in [26], a single self-attention layer is applied to perform global-level reasoning after analyzing the answer-aware entity graph, enabling the generation of the question token by token. A self-attention function can be described as mapping a query and a set of key–value pairs to an output, where the $\mathbf{Q}$, $\mathbf{K}$, and $\mathbf{V}$ come from the same vectors. The output is a weighted sum of the values, where the weights are computed by a compatibility function of the query with corresponding keys. With self-attention, $\mathbf{Q}$, $\mathbf{K}$, and $\mathbf{V}$, the interaction between words is investigated. This architecture is very useful for analyzing word associations between sentences by nature. The initial representations of $\mathbf{Q}$, $\mathbf{K}$, and $\mathbf{V}$ come from the obtained $M^P$. Then, $\mathbf{Q}$, $\mathbf{K}$, and $\mathbf{V}$ are processed as

$$\text{Attention}(\mathbf{Q}, \mathbf{K}, \mathbf{V}) = \text{softmax}(\frac{\mathbf{QK}_T}{\sqrt{d_k}})\mathbf{V}, \tag{14}$$

$$\text{head}_i = \text{Attention}(\mathbf{QW}_i^Q, \mathbf{KW}_i^K, \mathbf{VW}_i^V), \tag{15}$$

$$\text{MultiHead}(\mathbf{Q}, \mathbf{K}, \mathbf{V}) = \text{Cat}(\text{head}_1, \text{head}_2, \ldots, \text{head}_h)\mathbf{W}^o, \tag{16}$$

where $\mathbf{W}_i^Q$, $\mathbf{W}_i^K$, $\mathbf{W}_i^V$, and $\mathbf{W}^o$ are linear transformation matrices, and Cat indicates concatenation operation. In this work, $n = 8$ parallel attention heads are employed. For each of them, scaling factor $d_k$ is set to 64. Finally, a linear transformation and a softmax function are applied to convert the last token embedding of the multi-head self-attention to the next predicted token's probability.

## 4. Results
### 4.1. Dataset and Metrics

The performance of the proposed KGEL is evaluated on the HotpotQA dataset [28]. Unlike traditional QA tasks, where the question is given as input and the answer is expected as output, in HotpotQA, the input consists of the answer and context, and the system is expected to generate a question as output. The HotpotQA dataset is constructed using Wikipedia-based question–answer pairs that require multi-hop reasoning across multiple paragraphs. In our experiments, we filtered out questions that can be answered with a simple "yes" or "no" as these questions do not require multi-hop reasoning and thus have limited complexity. The resulting filtered dataset consists of 68k questions for training and 5k questions for testing. The evaluation is performed using BLEU, ROUGE-L, and METEOR metrics, using the nlg-eval package [29].

### 4.2. Baselines and Ablation Settings

In our comparative evaluation, we consider the several state-of-the-art models. Seq2Seq with Attention (Seq2Seq+Attn): a classic sequence-to-sequence architecture incorporating an attention mechanism for mapping passages to questions [30]. NQG++: an enhancement of the seq2seq model that includes an answer-aware input representation encoder incorporating answer position, POS, and NER information, which enables the generation of

answer-focused questions [19]. Answer-Separated Seq2Seq (ASs2s): a decoder of a question from an answer-separated passage encoder using a keyword net-based answer encoder to better capture key information in the target answer and generate appropriate questions [31]. S2sa-at-mp-gsa: an enhanced seq2seq model incorporating gated self-attention and maxout-pointers to address the challenge of processing long sentences in question generation (QG) [6]. GPT-2: a self-attention-based generation model capable of tackling tasks such as QA, machine translation, reading comprehension, and summarization. This model serves as a baseline for comparison with the proposed model [14].

Ablation settings for KGEL are defined as following. KGEL, which is the proposed knowledge graph-enhanced language model. KGEL-AT, which is KGEL without answer tagging (AT); it is the bi-attention component in Figure 2. KGEL-MHAR, which is KGEL without MHAR; it is the multi-head attention reasoning component in Figure 2. KGEL-EL, which is KGEL without evidence locating (EL); it is the answer-aware GAT in Figure 2.

It should be emphasized that the ablation experiment most relevant to this paper is KGEL-EL. We exclude the entire graph neural network, which is equivalent to the model we proposed without its key component: utilizing the graph to incorporate entity information. The model we employ here is a GPT-2 network with an augmented number of model layers to ensure a fair comparison, given that our network has more layers than GPT-2. It is important to note that our enhancement is not solely based on the inclusion of an extra layer, but rather the GNN has showcased an impressive capacity to merge logical reasoning.

**Implementation Details**. In the evaluation of the proposed KGEL model, open-source implementations of ASs2s and CGC-QG are employed on the HotpotQA dataset. The NQG++, S2sa-at-mp-gsa, and CGC-QG models utilize a 1-layer GRU as both the context encoder and question decoder, each with 512 hidden units. The word representations are 300-dimensional GloVe pre-trained embeddings. The baselines are optimized using Adam with a mini-batch size of 32. For the KGEL model, AdamW is used as the optimizer with a mini-batch size of 10 and a learning rate of from $3 \times 10^{-3}$ to $4 \times 10^{-3}$. The warm-up steps occupy 10% of the total training steps. The graph module of the KGEL model has two layers. The remaining experimental settings are kept consistent with the configuration in Radford et al. [14]. Hyperparameters of KGEL are listed in Table 1.

**Table 1.** Detailed hyperparameters of the model.

| Parameters | Values |
| --- | --- |
| Learning epoch | 50 |
| Dropout | 0.1 |
| Optimizer | AdamW |
| Batch size | 32 |
| Learning rate | $3 \times 10^{-3}$ to $4 \times 10^{-3}$ |
| Warm up | reach max lr at 10 epoch |
| Heads in GPT-2 | 12 |
| Layers of GPT-2 | 12 |
| Layers of GAT | 2 |
| Embeddding size | 768 |

### 4.3. Comparison with Baselines

The performance of all QG models on the HotpotQA dataset is presented in Table 2. The proposed model, KGEL, demonstrates superior performance compared to the baseline models across all evaluation metrics. Interestingly, the GPT-2 model outperforms the seq2seq-based models, which may be attributed to its larger size and the pre-training that captured general linguistic rules and knowledge. However, the KGEL model improves upon the GPT-2 model by achieving a higher score on BLEU-4, METEOR, and ROUGE-L, respectively, with an increase of 26.9%, 17.3%, and 10.0%. This result affirms the effectiveness of incorporating a knowledge graph and pre-trained language model in multi-hop

QG modeling. The contribution of each component to the improvement will be further analyzed in the ablation study.

**Table 2.** Evaluation results of different models by BLEU 1–4(B1-4),ROUGE-L(R-L), and ME-TEOR(M).The larger the number in the table, the better the result, with the bolded result representing the best result in the comparison model.

|                | B1    | B2    | B3    | B4    | R-L   | M     |
|----------------|-------|-------|-------|-------|-------|-------|
| seq2seq+Attn   | 32.97 | 21.11 | 15.41 | 11.81 | 18.19 | 33.48 |
| NQG++          | 35.51 | 22.12 | 15.53 | 11.50 | 16.96 | 32.01 |
| ASs2s          | 34.60 | 22.77 | 15.21 | 11.29 | 16.78 | 32.88 |
| S2sa-at-mp-gsa | 35.36 | 22.38 | 15.88 | 11.85 | 17.63 | 33.02 |
| GPT-2          | 39.00 | 24.46 | 17.21 | 12.71 | 16.81 | 32.06 |
| KGEL           | **41.93** | **28.04** | **20.83** | **16.13** | **19.70** | **35.28** |

*4.4. Ablation Tests Analysis*

The results of the three ablation tests conducted to evaluate the impact of the AT, multi-head attention, and EL modules in KGEL are presented in Table 3. The incorporation of answer information, indicating whether a word is within or outside the target answer, has been shown to play a critical role in seq2seq-based models on single-hop QG tasks [6,8,10]. This information has been included in almost all Seq2Seq-based QG models [6,8,10]. An ablation study conducted on the S2s-at-mp-gsa model showed that the AT significantly improves QG performance, highlighting the importance of providing the model with the information of what to ask [6].

**Table 3.** Evaluation results of ablation tests.The larger the number in the table, the better the result, with the bolded result representing the best result.

| Model    | B1    | B2    | B3    | B4    | R-L   | M     |
|----------|-------|-------|-------|-------|-------|-------|
| KGEL     | **41.93** | **28.04** | **20.83** | **16.13** | 19.70 | 35.28 |
| KGEL-AT  | 41.14 | 27.32 | 20.00 | 15.24 | **20.12** | **35.53** |
| KGEL-MHA | 38.84 | 25.00 | 18.03 | 13.66 | 17.46 | 32.98 |
| KGEL-EL  | 39.13 | 25.10 | 18.11 | 12.86 | 16.41 | 32.76 |

In the absence of answer information, the performance of KGEL-AT in terms of BLEU decreases, whereas the performance in terms of METEOR and ROUGE-L slightly increases, making the results of these two models comparable. This difference in performance may be due to the relative simplicity of single-hop QG tasks, which do not require multi-step reasoning, as opposed to multi-hop QG. The significance of answer information in this task is therefore more pronounced. In contrast, answer information is less significant in multi-hop QG, as this task requires reasoning between multiple contents, a challenge addressed by the proposed graph attention mechanism.

The results of the ablation tests evaluating the impact of multi-head attention (MHA) and evidence locating (EL) modules on the KGEL model are presented in Table 3. The performance of the KGEL model is significantly impacted by the removal of the multi-head attention mechanism, with reductions of 2.46, 2.24, and 2.3 in BLEU-4, METEOR, and ROUGE-L scores, respectively. This result supports the significance of multi-head attention in enabling the interaction across the fused representation of the input text to be modeled by the KGEL. Similarly, the removal of the EL module results in a significant decrease in performance, with KGEL outperforming KGEL-EL by 25.4%, 20.0%, and 7.7% in terms of BLEU-4, METEOR, and ROUGE-L, respectively. This highlights the effectiveness of the evidence locating component in establishing the reasoning chain between the answer and answer-related entities, which is critical for multi-hop QG.

*4.5. Case Study*

In order to better understand the generation of GPT-2, KGEL, and human annotations, a case study is conducted. The study presents four examples, two of which are selected from the positive generation and two from the negative generation. The negative generation produced by KGEL is demonstrated in Tables 4 and 5, whereas the positive generation is depicted in Tables 6 and 7. The case study provides insight into the strengths and weaknesses of the models and helps to evaluate the quality of the generated questions.

We further evaluate the performance of KGEL through a human evaluation of 150 randomly selected test samples. The evaluation scores are given as integers on a scale of 1 to 5, with a higher score indicating better performance. Three raters were asked to evaluate the generated questions from GPT-2, KGEL, and ground truth based on three metrics: fluency, answerability, and completeness. Fluency assesses the logical and grammatical correctness of the generated questions. Answerability assesses the relevancy and answerability of the generated questions in relation to the context. Completeness assesses the ability of the generated questions to involve multi-hop reasoning across scattered evidence in the context. The results of the human evaluation are shown in Table 8. Our results show a significant improvement over the baseline model, as demonstrated by the following three observations.

- The high fluency scores obtained by both KGEL and GPT-2 with a pre-trained language model highlight the ability of these models to generate grammatically correct and logical questions. However, the lower answerability and completeness scores of GPT-2 compared to human performance indicate its limitations in identifying key information from the context and determining "what to ask".
- The improved completeness score of KGEL compared to GPT-2 highlights the effectiveness of our proposed reasoning module in recognizing multi-hop relationships and identifying relevant entities. This demonstrates the better capability of our model in generating questions with multi-hop relations.
- The better performance of KGEL in terms of answerability, compared to GPT-2, is a result of the incorporation of entity graph in our reasoning module. However, further improvement is required in the ability of the model to identify the correct target aspect to ask, in order to enhance answerability.

As shown in Table 4, it is shown that both GPT-2 and KGEL get the wrong attention for the entity to ask. One challenge is that the support facts do not have any clue about "Gary L. Bennett".

In Table 5, both GPT-2 and KGEL ask a wrong question, unable to decide what to ask. The generation of KGEL is slightly better because the generation is more readable.

**Table 4.** Negative example 1.

| | |
|---|---|
| **Support Fact A:** | Ulysses is a decommissioned robotic space probe whose primary mission was to orbit the Sun and study it at all latitudes. |
| **Support Fact B:** | His professional career has included work on the Voyager, Galileo, and Ulysses space missions, and is currently working as a consultant in aerospace power and propulsion systems. |
| **Answer:** | Sun |
| **GPT-2:** | What is the name of the support space of the space named after the physicist who has work on the voyager? |
| **KGEL:** | David B. Mcdonald worked on the casting of the decommissioned spacecraft that are setting what landform? |
| **Ground Truth:** | Gary L. Bennett was a part of the space missions that have a primary destination of what celestial body? |

**Table 5.** Negative example 2.

| Support Fact A: Support Fact B: | In 2000, "Linux Format" called Qvwm "an unusually impressive imposter". Linux Format was the UK's first Linux-specific magazine, and as of 2013 was the best-selling Linux title in the UK. |
|---|---|
| Answer: | An unusually impressive imposter |
| GPT-2: KGEL: Ground Truth: | What was the first Linux-such first linux-specific magazine? What type of magazine called qvwm "an unusually impressive imposter"? What did the UK's first Linux-specific magazine call Qvwm? |

In Table 6, KGEL generates a variant of a golden question with a similar meaning as the human annotation, whereas the GPT-2 generates inexplicable words. The presented example serves as an exemplary solution to the primary challenge of accurately identifying critical evidence. Through the optimization of our framework, the model successfully identified the relevant question direction of "SPACE system business" in contrast to the erroneous response of "Nobleman" produced by GPT-2. This achievement underscores the effectiveness of our designed framework in identifying and selecting critical evidence with a high degree of accuracy.

**Table 6.** Positive example 1.

| Support Fact A: Support Fact B: | SpaceDev, a part of the "Space Systems Business" of Sierra Nevada Corporation, is prominent for its spaceflight and microsatellite work. Sierra Nevada Corporation operates under the leadership of Chief Executive Officer, Fatih Ozmen and President, Eren Ozmen. |
|---|---|
| Answer: | Fatih Ozmen |
| GPT-2: KGEL: Ground Truth: | Who is the president of the company that operates Russian Li: Nobleman? Who is the president of the corporation that is a member of the space systems business? Who is the Chief Executive Officer of the corporation that owns Space Dev? |

As shown in Table 7, KGEL generates a valid question for the answer, whereas GPT-2 generates a question with the wrong semantic relationship. This example is a compelling illustration, and it is imperative to highlight that it was selected randomly to accentuate the robustness of our model. The problem presented is a typical multi-hop challenge that involves intricate and diverse entity types, including time, personal names, workplace, nationality, and publishing houses. Such complexities necessitate sophisticated and intricate reasoning. Notably, our model, although phrased differently from the original question, was able to capture the essence of the problem. Conversely, the GPT-2 model produced paragraphs that were linguistically sound but devoid of any logical coherence. This underscores the pre-training model's capacity for linguistic fluency but indicates its inadequacy in complex reasoning tasks.

Reviewing the generation of KGEL produces some thoughts for future direction:

- The generation quality of multi-hop questions is far from desired;
- Incorporating external knowledge may help the model recognize the relationship between entities and answers;
- A copy mechanism has the potential to be introduced to help the model generate the shared semantic content in the target question for many words in question that come from context.

**Table 7.** Positive example 2.

| | |
|---|---|
| **Support Fact A:** | Anne of Ingleside is a children's novel by Canadian author Lucy Maud Montgomery. |
| | It was first published in July 1939 by McClelland and Stewart (Toronto) and the Frederick A. Stokes Company (New York). |
| **Support Fact B:** | Lucy Maud Montgomery '1' :, '2':, '3':, '4':  (30 November 1874–24 April 1942) published as L.M. Montgomery, was a Canadian author best known for a series of novels beginning in 1908 with "Anne of Green Gables" |
| | The central character, Anne Shirley, an orphaned girl, made Montgomery famous in her lifetime and gave her an international following. |
| **Answer:** | Lucy Maud Montgomery |
| **GPT-2:** | Who wrote the central character's novel in which the fictional female protagonist of the novel was first published in July 1939? |
| **KGEL:** | Who is the author best known for a series of novels about Anne Shirley which was first published in July 1939, by Mcclelland and Stewart? |
| **Ground Truth:** | Wich children's novelist whow was first published in 1939 gained an internaional following writting about an orphaned girl named Anne Shirley? |

**Table 8.** The results of human evaluation.

| Model | Fluency | Answerability | Completeness |
|---|---|---|---|
| GPT-2 | 3.85 ($\pm$0.30) | 2.14 ($\pm$0.97) | 2.68 ($\pm$0.71) |
| KGEL | 4.43 ($\pm$0.41) | 2.92 ($\pm$0.76) | 3.28 ($\pm$0.34) |
| Human | 4.87 ($\pm$0.13) | 4.96 ($\pm$0.21) | 4.86 ($\pm$0.26) |

## 5. Conclusions

The present study proposes a novel approach to question generation, which leverages the benefits of a knowledge graph and mimics human reasoning. The proposed model, referred to as KGEL, incorporates an answer-aware graph reasoning module to improve the ability to identify key information from the context. Empirical evaluations on the HotpotQA dataset demonstrate the superiority of KGEL over baseline models, particularly in terms of the completeness and answerability of generated questions.

In conclusion, this paper provides a promising approach for multi-hop question generation and highlights the potential for further improvement. Integrating external knowledge and incorporating a copy mechanism are identified as promising avenues for future research. These directions aim to enhance the ability of the model to recognize relationships across semantic content and generate shared semantic information in the target question.

The results obtained from our experiment demonstrate relatively good performance compared to that of GPT-2 fine-tune. However, we must consider that our training data are limited to the Hotpot QA dataset. Given recent demonstrations of ChatGPT's remarkable prowess and the sophisticated techniques employed therein, it is improbable that our model could achieve the complexity required for multi-hop question answering inference using only the Hotpot QA dataset. In contrast, ChatGPT's exceptional performance highlights the potential limitations of our model arising from insufficient data, and prompt training on more pertinent data would likely result in a more potent model.

**Author Contributions:** Z.L.: software, methodology, writing—original draft; Z.C.: methodology, writing—review and editing, validation; P.L.: investigation, writing—review and editing, validation; Y.Z.: writing—review and editing, supervision; S.L.: writing—review and editing, supervision, project administration. All authors have read and agreed to the published version of the manuscript.

**Funding:** This work was supported by the AI industrial technology innovation platform of Sichuan Province, grant number "2020ZHCG0002".

**Institutional Review Board Statement:** Not applicable.

**Informed Consent Statement:** Not applicable.

**Data Availability Statement:** The source code is available at https://github.com/zhenpingli/KGEL, accessed on 1 May 2023.

**Conflicts of Interest:** The authors declare no conflict of interest.

## Abbreviations

The following abbreviations are used in this manuscript:

| | |
|---|---|
| MDPI | Multidisciplinary Digital Publishing Institute |
| DOAJ | Directory of open access journals |
| TLA | Three letter acronym |
| LD | Linear dichroism |

## Appendix A. The Illustrator of Data Processing

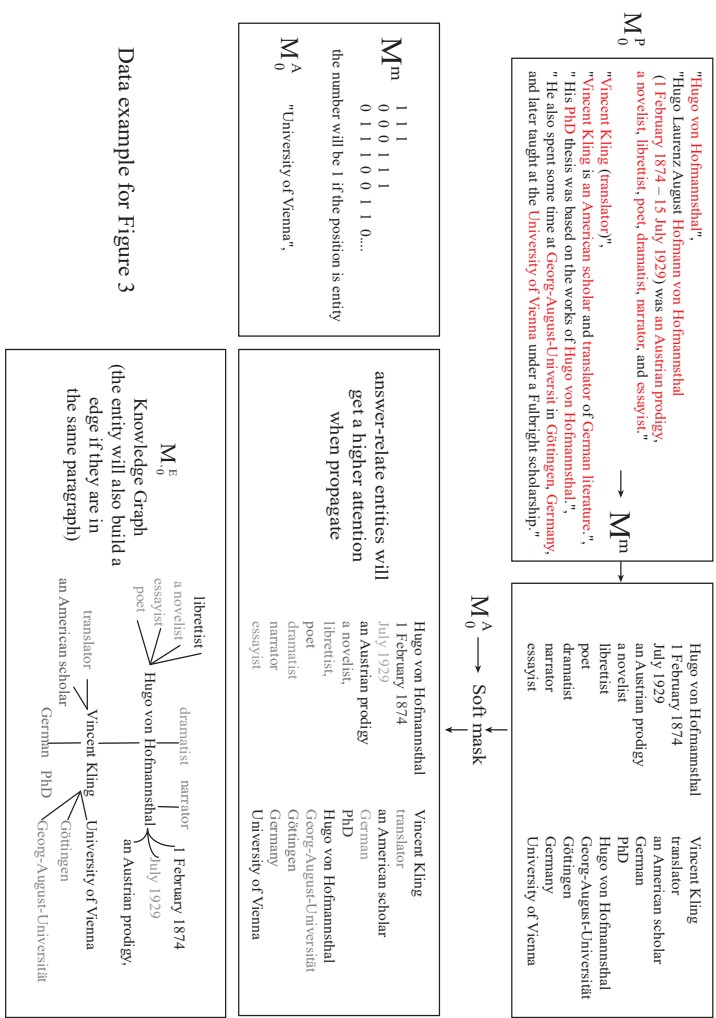

**Figure A1.** Data example of Figure 3.

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
