# Peer review of "Multi-Hop Question Generation with Knowledge Graph-Enhanced Language Model"

_applsci, doi:10.3390/app13095765_

Round 1
Reviewer 1 Report
The paper presents a method for generating questions requiring multiple reasoning steps. The proposed neural network architecture, KGEL, is based on transformers (GPT-2) and graph neural networks (GAT). The proposed KGEL is evaluated using the HotpotQA dataset (multi-hop QA) using a set of standard language generation metrics and human evaluation. The results show that KGEL outperforms other works in all metrics but does not reach human performance. Additional ablation analysis is provided to justify the components of the proposed architecture.
Concerns
1. Since the paper proposed the KGEL "to cope with the challenges," it was essential that the authors showed how the challenges were mitigated. The human evaluation is good evidence to discuss this.
2. The description of the method is not entirely clear. The paper initially described the three components, but there seems to be little connection with the rest of the section. While individual subsections of section 3 are readable, there is no connection between them and the challenges or the components. In addition, it is unclear why MA from Eq 12 is needed. Similarly, in the ablation analysis, the "-AT", "-MHA", and "-EL" do not link to the main description of the model.
3. The language generation metrics show strong results. However, in the human evaluation, KGEL does not seem to be statistically significant from GPT2. The author should use more samples and compute annotator agreement along with the significant test.
Author Response
Dear Reviewer:
Thank you for taking the time to review my work and pointing out the mistake.
- Since the paper proposed the KGEL "to cope with the challenges," it was essential that the authors showed how the challenges were mitigated. The human evaluation is good evidence to discuss this.
I agree with you very much. This is an oversight of my paper writing. In fact, it is a very good example in Table7.
The problem presented is a typical multi-hop challenge that involves intricate and diverse entity types, including time, personal names, workplace, nationality, and publishing houses. Such complexities necessitate sophisticated and intricate reasoning. Notably, our model, although phrased differently from the original question, was able to capture the essence of the problem. Conversely, the GPT-2 model produced paragraphs that were linguistically sound but devoid of any logical coherence. This underscores the pre-training model's capacity for linguistic fluency but indicates its inadequacy in complex reasoning tasks.
Table 6 presents a relatively less challenging set of problems, where entities must be interconnected before any logical reasoning can take place. GPT-2, as observed from the results, performed poorly, and provided incorrect answers. However, after optimization with graph neural networks, our model was able to capture critical information and question directions more accurately, thereby improving performance. This underscores the effectiveness of our framework in addressing complex reasoning challenges that require the accurate interconnection of entities.
- 2. The description of the method is not entirely clear. The paper initially described the three components, but there seems to be little connection with the rest of the section. While individual subsections of section 3 are readable, there is no connection between them and the challenges or the components. In addition, it is unclear why MAfrom Eq 12 is needed. Similarly, in the ablation analysis, the "-AT", "-MHA", and "-EL" do not link to the main description of the model.
Your suggestion was excellent, and I successfully incorporated it into the article by linking the ablation experiment with the presented challenge, resulting in a clearer and more cohesive paragraph. I appreciate your valuable input, which has greatly contributed to the quality of the article.
KGEL-AT which is KGEL without answer tagging (AT), it is the Bi-attention component in Figure 1.
KGEL-MHAR which is KGEL without MHAR, it is the multi-head attention reasoning component in Figure 1.
KGEL-EL which is KGEL without evidence locating (EL), it is the Answer-aware GAT in Figure 1.
- The language generation metrics show strong results. However, in the human evaluation, KGEL does not seem to be statistically significant from GPT2. The author should use more samples and compute annotator agreement along with the significant test.
In all honesty, while there has been a significant improvement in the machine score in comparison to GPT-2, as highlighted in my case study like Table 7, the experiment has yielded some remarkable reasoning outcomes. However, in comparison to the impressive results achieved by the recent powerful chatGPT, the quality of multi-hop question generation falls short of expectations. We are presently endeavoring to further explore chatGPT's performance, but the endeavor is proving challenging due to the hardware requirements and the enormity of the data set needed to train the model.
We have released the entire codebase and uploaded all test results data. However, it is unclear whether the current inability to view the complete code on our GitHub repository is due to review regulations.
We hope that the revised version of the paper meets your expectations.

Reviewer 2 Report
Dear Sir
I have no comments about this research paper due to seem that everything is organized and technically sound. Moreover, I suggest the authors to check English grammar and linguistic mistakes in order to make sure this paper is free of mistakes, since then it will be ready to publish it.
Good Luck
Good Luck
Author Response
Dear Reviewer:
Thank you for taking the time to review my work and pointing out the mistake.
I have made revisions to clarify the language and have also addressed any grammatical errors that were present. Your feedback has been invaluable, and I greatly appreciate your input.
We hope that the revised version of the paper meets your expectations.

Reviewer 3 Report
In this manuscript, the authors proposed a NLP model to question generation based on knowledge graph that called KGEL. The research topic and structure is interesting. However I would like to suggest a few comments:
- The paper should briefly mention the limitations of the conducted research.
- The description about implementing/settings in the experiment section can be better.
- In the section 3, that would be better if provide a pseudocode for KGEL to more understand the research model for readers.
Author Response
Dear Reviewer:
- The paper should briefly mention the limitations of the conducted research.
I added a discussion of the limitations of the algorithm at the end of the article.
Given recent demonstrations of ChatGPT's remarkable prowess and the sophisticated techniques employed therein, it is improbable that our model could achieve the complexity required for multi-hop question answering inference using only the Hotpot QA dataset. By contrast, ChatGPT's exceptional performance highlights the potential limitations of our model arising from insufficient data, and prompt training on more pertinent data would likely result in a more potent model.
- 2. The description about implementing/settings in the experiment section can be better.
I list the details of network’s setting in table 2.
- In the section 3, that would be better if provide a pseudocode for KGEL to more understand the research model for readers.
We have released the entire codebase and uploaded all test results data. However, it is unclear whether the current inability to view the complete code on our GitHub repository is due to review regulations.
We hope that the revised version of the paper meets your expectations.
